# UPS: Unified Projection Sharing for Lightweight Single-Image Super-resolution and Beyond

**Kun Zhou**[1,2]*, **Xinyu Lin**[1,2]†, **Zhonghang Liu**[3], **Xiaoguang Han**[1]‡, **Jiangbo Lu**[2]‡

[1]SSE, CUHK-Shenzhen,    [2]SmartMore Corporation    [3]SMU, Singapore

hanxiaoguang@cuhk.edu.cn,  jiangbo.lu@gmail.com

## Abstract

To date, Transformer-based frameworks have demonstrated impressive results in single-image super-resolution (SISR). However, under practical *lightweight* scenarios, the complex interaction of deep image feature extraction and similarity modeling limits the performance of these methods, since they require simultaneous *layer-specific* optimization of both two tasks. In this work, we introduce a novel Unified Projection Sharing (UPS) algorithm to decouple the feature extraction and similarity modeling. To achieve this, we establish a unified projection space defined by a learnable projection matrix, for similarity calculation across *all* self-attention layers. As a result, deep image feature extraction remains a per-layer optimization manner, while similarity modeling is carried out by projecting these image features onto the shared projection space. Extensive experiments demonstrate that our proposed UPS achieves state-of-the-art performance relative to leading lightweight SISR methods, as verified by various popular benchmarks. Moreover, our unified optimized projection space exhibits encouraging robustness performance for unseen data (degraded and depth images). Finally, UPS also demonstrates promising results across various image restoration tasks, including real-world and classic SISR, image denoising, and image deblocking.

## 1 Introduction

Single-image super-resolution is a fundamental task in computer vision, aiming to enhance the resolution and quality of a low-resolution image. Recently, Transformer-based methods [1–6], especially, SwinIR [7], combines the benefits of window-based self-attention and convolutional feature extraction, thus achieving effective similarity modeling and feature extraction. It yields promising outcomes, reducing computational demand compared to global/non-local attention mechanisms.

However, the coupled optimization in existing Transformer-based methods may face two challenges. First, in a lightweight configuration characterized by a very limited number of learnable parameters, performing layer-specific optimization for both image feature extraction and similarity modeling remains challenging. Second, such a tightly coupled optimization scheme (image feature extraction and projection similarity are synchronously updating in each layer during the training phase) may suffer from co-adaptation issue [8, 9], potentially leading to inferior results.

Interestingly, we observe that projection spaces (layer) in trained SwinIR-light exhibit *substantial* layer-to-layer (CKA [10]) similarities[4]. Fig. a.(1-3) below shows over 0.95 (0.99, 0.95, 0.96) for

---

*Project leader

†Co-first author

‡Corresponding author

[4]The dimensions remain consistent across all projection layers in SwinIR-light. Thus we can directly evaluate the pair-wise similarity scores.

38th Conference on Neural Information Processing Systems (NeurIPS 2024).

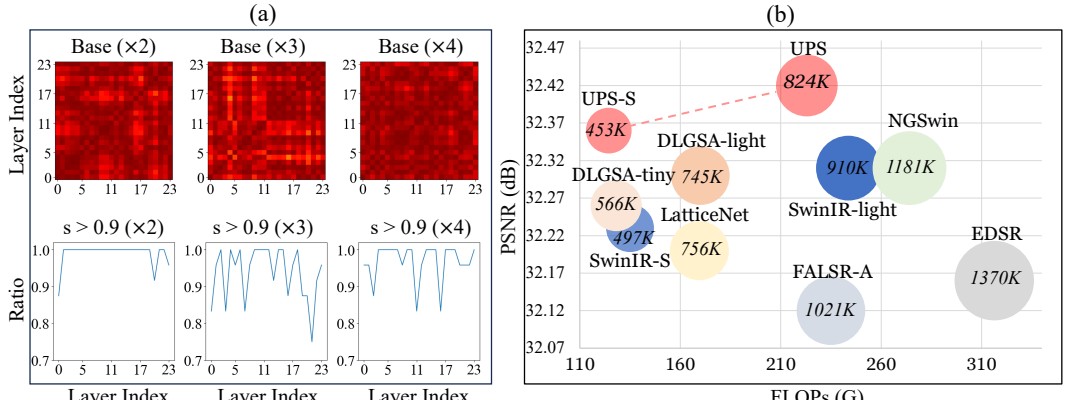

Figure 1: (a) We observe that the SwinIR-light (termed as Base) models exhibit significant similarities (CKA [10]) in projection layers. (b) Comparison between our proposed UPS and SOTA lightweight SISR models on BSD100 [11] for ×2 setting. A bigger circle size means a larger number of parameters. While being the most computationally and parameter-efficient, UPS-S (a more lightweight version of our method) demonstrates highly competitive results compared to SOTA methods.

×{2, 3, 4}) (projection layer) pairs get over 0.9 scores (ranging from 0 to 1)[5]. This experiment suggests that all the projection layers are *highly similar*.

To mitigate the two problems, we are motivated by the observation and explore a novel Unified Projection Sharing (UPS) technique for lightweight SISR. In particular, UPS decouples the deep image feature representation and similarity learning: it performs the layer-specific image feature extraction while calculating the self-similarity in a unified projection space. In other words, the similarity modeling is optimized in a *layer-invariant* manner, effectively separating the learning of both two tasks. More specifically, UPS accomplishes self-similarity modeling with the following three steps: (i) UPS defines a unified projection space by a learnable matrix; (ii) for each self-attention layer, it projects deep image features onto the unified projection space; (iii) it calculates the attention map using the Cosine similarity metric in the projection space and performs attention-based aggregation.

Our proposed UPS consistently demonstrates superior performance compared to existing approaches across all testing benchmarks. Notably, our method outperforms the second-best model by more than 0.33dB on the Manga109 dataset for the ×2 settings. Furthermore, our model exhibits significant improvement over our baseline model, SwinIR-light [7], achieving enhancements of up to 0.50dB, 0.55dB, 0.47dB on the Manga109 dataset for the ×2, ×3, ×4 settings, while utilizing fewer parameters. Our contributions are summarized as follows:

− We propose UPS, an effective decoupled SISR optimization framework, to address the challenge of simultaneous layer-specific feature extraction and similarity modeling for lightweight SISR.

− UPS simplifies the similarity optimization process by learning a layer-invariant projection space, leading to effective aggregation (activating more local/non-local pixels as shown in Fig. 3) and improved performance, even with reduced model capacity (see Fig. 1) and less training samples (see the data efficiency analysis in Sec. A.2).

− Extensive robustness analysis in Sec. 5.4, 5.5, A.3, A.4, have confirmed the good generalization ability of our proposed UPS for unseen data, such as noisy image and depth map SR.

## 2   Related Works

**CNN-based SISR.** Due to their low complexity and helpful feature extraction abilities, CNNs have been widely used for SISR task. SRCNN [12] pioneered the use of deep convolutional neural network (CNN) architectures specifically designed for single image super-resolution (SISR). SRCNN consists of only three layers: patch extraction, non-linear mapping, and reconstruction. It has demonstrated competitive performance compared to traditional non-deep methods, inspiring the development of numerous lightweight CNN approaches in the SISR field. ESPCN [13] introduced a compact network

---

[5]The numerical values on the axes indicate the layer indices.

architecture that employs sub-pixel convolutional layers to upscale low-resolution image features. In contrast, LapSRN [14] utilizes image structure priors across different pyramid representations, resulting in improved performance while minimizing computational overhead. Taking inspiration from dictionary-learning models [15–17], LAPAR [18] learns linear coefficients associated with pre-defined basic up-sampling kernels to produce an optimal pixel-specific kernel, achieving superior super-resolution results. LatticeNet [19] designs a parameter-efficient convolutional lattice block to extract hierarchical contextual features. Despite their computational efficiency, CNN-based models are limited in terms of long-term aggregation due to content-invariant similarity optimization.

**Transformer-based SISR.** Recently, Transformer-based techniques [20–25] have achieved remarkable outcomes in SISR but still suffer from high complexity. The computational cost of non-local self-similarity modeling increases quadratically with the size of the image. Inspired by the success of Swin Transformer [26], numerous window-based Transformer frameworks emerged to address the efficiency of SISR. For example, SwinIR [7] introduces a residual window-based transformer block (RSTB) for image feature extraction and similarity-based aggregation, outperforming previous CNN-based and Transformer-based approaches. DLGSA-l [27] proposes a global sparse attention technique to enhance the aggregation of relevant tokens. NGswin [28] incorporates the N-Gram context to attain a larger receptive field, activating more neighboring pixels for effective aggregation. However, when it comes to lightweight setups, the optimization of coupled feature extraction and similarity calculation is limited, resulting in inferior performance.

**Efficient Transformers.** On the other hand, some advanced transformers have been proposed to reduce the computational complexity, enhancing inference or training efficiency. ShareFormer [29] presents a local similarity map-sharing scheme between neighboring attention layers for lower latency. Thus, ShareFormer shares a static similarity map for neighboring attention layers while UPS calculates dynamic similarity maps with layer-refined features in a shared projection space.

Skip-Attention [30] cuts off some intermediate attention layers to improve efficiency and performance for high-level tasks. LaViT [31] proposes a residual-based attention downsampling that fuses the initial calculated attention scores to guide the aggregation of the following layers, resulting in faster efficiency and improved classification accuracy.

Therefore, Skip-Attention and LaViT follow the existing coupled optimization scheme (reduce some attention calculations), and UPS proposes a decoupled learning strategy to enhance performance. We will cite the insightful studies and add this discussion to our revised paper.

# 3   Understanding Swin Transformer

**Preliminaries.** Swin Transformer [26] proposes an effective self-attention mechanism, achieving long-range information capture at a lower computation complexity. Inspired by Swin Transformer, several subsequent methods [7, 32, 28] dedicated to solving lightweight SISR have emerged, consistently enhancing the quality of super-resolved images. Fig. 2(a) illustrates the general framework architecture of the Swin Transformer-based SISR method. It consists of three primary components: a shallow head module, a deep image feature extraction and aggregation (FEA) module, and a tail reconstruction module. The head module is tasked with converting the input low-resolution RGB image into a high-dimensional feature space. The FEA module, the key role in the whole architecture, is composed of multiple ($N$) Swin Transformer layers (STLs). Each STL has two main objectives: (i) extracting image features and (ii) modeling similarities using a learnable projection space. The former focuses on capturing essential image features, while the latter employs window-based self-attention to facilitate spatially adaptive aggregation. Notably, similarity modeling optimizes a projection space to obtain pixel-wise correlations, which is achieved by projecting image features into the learnable projection space and calculating similarity scores. Finally, the tail module generates the final high-resolution output image, completing the SISR process. In the subsequent section, we will delve into the details of the STL, with a particular emphasis on deep feature extraction and similarity modeling aspects.

## 3.1   Decomposing Swin Transformer Layer

Efficient deep feature extraction and similarity modeling are accomplished by the Swin Transformer Layer (STL), the fundamental unit within the FEA module of the Swin Transformer. Illustrated in

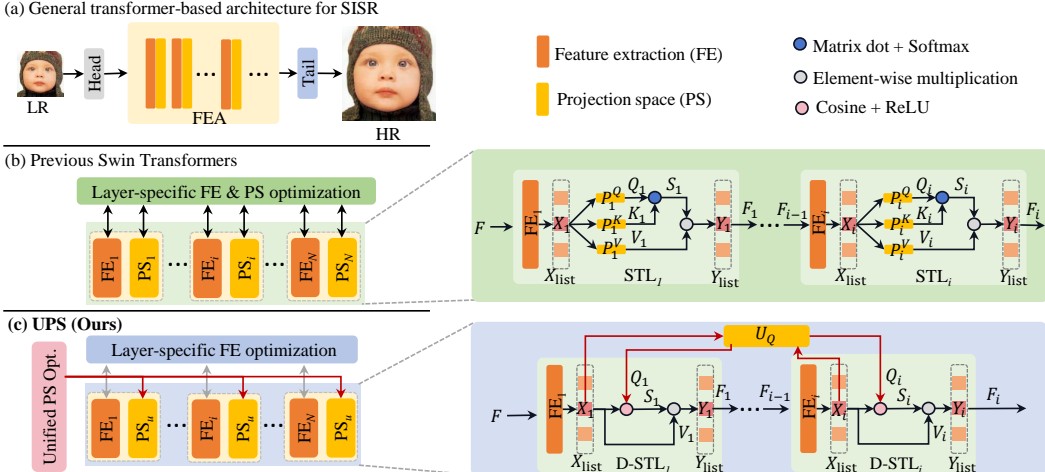

Figure 2: Overview of Transformer-based architecture for lightweight SISR. There are three main components: (i) a head shallow feature extraction module, (ii) a deep feature extraction and aggregation (FEA) module consisting of $N$ Swin Transformer layers ($STL_1,\cdots,STL_N$), and (iii) a tail reconstruction module. Previous transformers (i.e., SwinIR [7], NGSwin [28]) synchronously perform multiple layer-specific deep image feature extraction (FE) and projection space (PS) optimization within a Swin Transformer Layer (STL). In contrast, we develop a decoupled Swin Transformer Layer (D-STL) in UPS to optimize per-layer feature extraction and a unified projection space ("$PS_u$" defined by a learnable projection matrix $U^Q$).

Fig. 2(b), in the $i$-th STL, the process begins by employing a convolutional layer to extract deep image feature $\hat{F}_i$ from an input feature $F_{i-1}$, which is the output of the preceding $(i-1)$-th STL:

$$\hat{F}_i = \text{Conv}(F_{i-1}). \tag{1}$$

Subsequently, the STL executes a conventional window-based self-attention mechanism, comprising four basic steps: (i) window-partitioning, (ii) deep feature projection for similarity calculation, (iii) aggregation based on similarity to merge neighboring pixels, and (iv) patch merging.

(i) **Window-partitioning.** Initially, the updated image feature $\hat{F}_i$ is reshaped into $\frac{HW}{M^2}$ non-overlapping patches, each with a shape of $M^2 \times C$, where $M^2$ represents the spatial size of each patch and $C$ is the channel dimension.

(ii) **Layer-specific projection.** Following window-partitioning, each divided image patch $X_i$ from $\hat{F}_i$ is projected to generate the corresponding query, key, and value matrices $Q_i, K_i, V_i$:

$$Q_i = X_i P_i^Q, \quad K_i = X_i P_i^K, \quad V_i = X_i P_i^V, \tag{2}$$

where $P_i^Q, P_i^K, P_i^V \in R^{\{d \times C\}}$ denote the learnable projection parameters specific to the $i$-th STL, while $Q_i, K_i, V_i \in R^{\{M^2 \times d\}}$ represent the projected features of patch $X_i$ and $d$ is the projection dimension. The similarity matrix is then computed:

$$S_i = \text{SoftMax}\left(\frac{Q_i K_i^T}{\sqrt{d}} + B_i\right), \tag{3}$$

where $B_i$ represents a relative position encoding, and $S_i$ is the predicted similarity map for the $X_i$.

(iii) **Similarity-based Aggregation.** Later on, neighboring information within the patch $X_i$ is aggregated based on the computed similarity map $S_i$:

$$Y_i = S_i V_i. \tag{4}$$

(iv) **Patch Merging.** Finally, all the aggregated image patches are reshaped into a 2D image feature which is fed into the next STL for further processing.

**Discussion.** With sufficient model capability, i.e., millions of parameters, SwinIR [7], a SOTA Swin Transformer SISR model, exhibits strong abilities for the SISR task. However, in resource-constrained, lightweight settings as previously mentioned, it potentially poses challenges to simultaneously optimize deep image feature extraction and projection space. We will compare the per-layer projection space optimization with our proposed UPS scheme later.

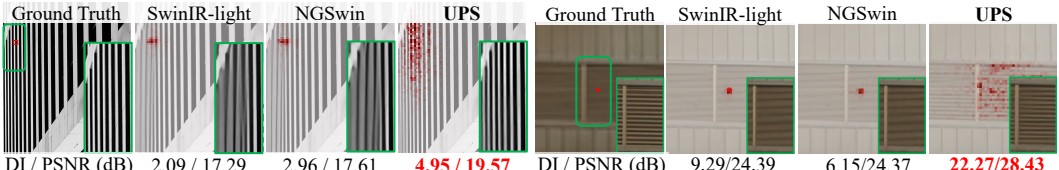

| | Ground Truth | SwinIR-light | NGSwin | **UPS** | | Ground Truth | SwinIR-light | NGSwin | **UPS** |
|---|---|---|---|---|---|---|---|---|---|
| DI / PSNR (dB) | 2.09 / 17.29 | 2.96 / 17.61 | **4.95 / 19.57** | | DI / PSNR (dB) | 9.29/24.39 | 6.15/24.37 | **22.27/28.43** | |

Figure 3: Comparison between SOTA SISR models and ours. We show the SR results overlaid with the local attribution map (LAM [33]) of each model. The LAM visually illustrates the activation of local and non-local pixels involved in super-resolving the highlighted patch within the red box. The numbers beneath are the DI (↑) [33] and PSNR (↑) values. Zoom in for better visual comparison.

## 4 Unified Projection Sharing for Lightweight SISR

**Overview.** To address the entanglement optimization of image feature extraction and similarity modeling, we introduce a Unified Projection Sharing (UPS) technique for lightweight SISR. Fig. 2(b), (c) summarizes the optimization schemes of existing Transformer-based SISR frameworks and our proposed UPS. As can be seen, previous methods typically focus on jointly optimizing deep image features and similarity modeling within each layer. In contrast, UPS adopts a shared projection space for similarity modeling, allowing layer-specific feature extraction while separating the optimization of similarity calculation.

### 4.1 Unified Projection Sharing

We follow the general framework structure of Swin Transformer but use decoupled projection space optimization. As shown in Fig. 2(c), UPS consists of three basic modules, namely the convolutional head module, FEA module, and reconstruction tail module. In the FEA, we develop a decoupled Swin Transformer layer (D-STL) for deep image feature extraction, while optimizing a unified projection space for similarity modeling. Next, we will provide a detailed description of our D-STL.

### 4.2 Decoupled STL (D-STL)

We take the $i$-th D-STL for illustration. Given an input image feature $F_{i-1}$ produced by the last $(i-1)$-th D-STL, we aim to perform feature updating as well as self-similarity-based aggregation. Similarly, we adopt the Eq. 1 to conduct deep image feature extraction and obtain the transformed image feature $\hat{F}_i$. Then we employ the window-partitioning process to reshape the $\hat{F}$ into $\frac{HW}{M^2}$ non-lapped image patches.

**Unified Projection.** Unlike the layer-specific projection scheme in Swin Transformers, we introduce a layer-invariant (unified) projection space defined by a learnable matrix $U^Q \in R^{\{D \times C\}}$ ($D$ refers to the unified projection dimension) and project the deep feature $X_i$ on this unified projection space:

$$Q_i = X_i U^Q, \quad V_i = X_i. \tag{5}$$

After that, we consider the calculation of the self-similarity in the unified projection space. Motivated by ReLUFormer [34] that addresses the over-centralized distribution in Softmax by incorporating ReLU activation for self-similarity calculation, we get the similarity scores as:

$$S_i = \text{ReLU}(\text{Cosine}(Q_i, Q_i^T) + B_i). \tag{6}$$

Note that we conduct normalization operation for the projected image features $Q_i$ [6]. Subsequently, we utilize the Cosine similarity metric, followed by a ReLU activation function, to obtain the final similarity map $S_i$. We also assess our design in Sec. 5.3. Finally, leveraging the calculated similarity map $S_i$, we perform image feature aggregation using Eq. 4.

**Discussion.** In Algorithm. 1, 2, we provide side-by-side illustrations of standard STL and our D-STL and highlight the differences between the two methods. The STL in previous Swin Transformers learns the coupled projection spaces and deep image feature extraction. In contrast, by a unified projection optimization scheme, each of our D-STLs only focuses on the deep image feature extraction. It largely

---

[6]To decrease the model complexity, we set the $K_i$ to be identical to the $Q_i$, as the SISR task typically involves only one data modality.

**Algorithm 1** Pseudo Code of the $i$-th STL

1: **Require**: Input $F_{i-1}$, window size $M$
2: Feature extraction: $\hat{F}_i = \text{Conv}(F_{i-1})$
3: Partitioning: $X_i^{\text{list}} = \text{Partitioning}(\hat{F}_i, M)$
4: Define aggregated patch list: $Y_i^{\text{list}}$
5: **for** $X_i$ **in** $X_i^{\text{list}}$ **do**
6:     Projection: $Q_i, K_i, V_i = X_i P_i^Q, X_i P_i^K, X_i P_i^V$
7:     Similarity cal.: $S_i = \text{SoftMax}\left(\frac{Q_i K_i^T}{\sqrt{d}} + B_i\right)$
8:     Aggregation: $Y_i = S_i V_i$
9:     $Y_i^{\text{list}}.append(Y_i)$
10: **end for**
11: **return** $\text{Reshape}(Y_i^{\text{list}})$

**Algorithm 2** Pseudo Code of the $i$-th Decoupled STL

1: **Require**: $F_{i-1}$, $M$, unified projection matrix $U^Q$
2: Feature extraction: $\hat{F}_i = \text{Conv}(F_{i-1})$
3: Partitioning: $X_i^{\text{list}} = \text{Partitioning}(\hat{F}_i, M)$
4: Define aggregated patch list: $Y_i^{\text{list}}$
5: **for** $X_i$ **in** $X_i^{\text{list}}$ **do**
6:     Projection: $Q_i, V_i = X_i U^Q, X_i$
7:     Similarity cal.: $S_i = \text{ReLU}(\text{Cosine}(Q_i, Q_i^D) + B_i)$
8:     Aggregation: $Y_i = S_i V_i$
9:     $Y_i^{\text{list}}.append(Y_i)$
10: **end for**
11: **return** $\text{Reshape}(Y_i^{\text{list}})$

reduces the overall optimization complexity by learning the similarity modeling in a unified projection space throughout all D-STL layers. As shown in Fig. 3, compared with SOTA lightweight Swin Transformers, UPS activates more non-local pixels and restores correct fine-grain image structures.

Table 1: Quantitative comparison with SOTA lightweight SISR methods on multiple benchmark datasets. The best and second-best results on the default training setting (DIV2K) are highlighted in **red** and **blue**, respectively. The "+" indicates that the two methods are trained on the DF2K dataset. We use **bold** to highlight the lowest FLOPs of Transformer-based methods. All FLOPs (also in Tab. 2b, 3,4) are calculated with an output size of $1280 \times 720$.

| Method | Scale | Parameters (K) | FLOPs (G) | Set5 PSNR / SSIM | Set14 PSNR / SSIM | BSD100 PSNR / SSIM | Urban100 PSNR / SSIM | Manga109 PSNR / SSIM |
|---|---|---|---|---|---|---|---|---|
| IMDN | | 694 | 158.8 | 38.00 / 0.9605 | 33.63 / 0.9177 | 32.19 / 0.8996 | 32.17 / 0.9283 | 38.88 / 0.9774 |
| RFDN-L | | 626 | 145.8 | 38.08 / 0.9606 | 33.67 / 0.9190 | 32.18 / 0.8996 | 32.24 / 0.9290 | 38.95 / 0.9773 |
| SwinIR-light | | 910 | 244.4 | 38.14 / 0.9611 | 33.86 / 0.9206 | 32.31 / 0.9012 | 32.76 / 0.9340 | 39.12 / 0.9783 |
| DLGSA-light | ×2 | 745 | 170.0 | 38.20 / 0.9612 | 33.89 / 0.9203 | 32.30 / 0.9012 | 32.94 / 0.9355 | **39.29** / 0.9780 |
| Omni-SR | | 772 | 194.5 | **38.22** / 0.9613 | 33.98 / 0.9210 | **32.36** / 0.9020 | **33.05** / **0.9363** | 39.28 / 0.9784 |
| **UPS** | | 824 | **162.5** | **38.26** / **0.9642** | **34.16** / **0.9232** | **32.42** / **0.9031** | **33.08** / **0.9373** | **39.62** / **0.9800** |
| SwinIR-S | ×2 | **497** | **107.3** | 38.06 / 0.9603 | 33.80 / 0.9186 | 32.23 / 0.9006 | 32.24 / 0.9301 | 38.76 / 0.9778 |
| **UPS-S** | ×2 | **453** | **90.6** | 38.16 / **0.9638** | **34.00** / **0.9220** | **32.36** / **0.9023** | 32.79 / 0.9346 | 39.26 / **0.9790** |
| Omni-SR+ | ×2 | 772 | 194.5 | 38.29 / 0.9617 | 34.27 / 0.9238 | 32.41 / 0.9026 | 33.30 / 0.9386 | 39.53 / 0.9792 |
| **UPS+** | ×2 | 824 | 162.5 | 38.31 / 0.9643 | 34.37 / 0.9247 | 32.43 / 0.9032 | 33.34 / 0.9388 | 39.80 / 0.9802 |
| IMDN | | 703 | 71.5 | 34.36 / 0.9270 | 30.32 / 0.8417 | 29.09 / 0.8046 | 28.17 / 0.8519 | 33.61 / 0.9445 |
| RFDN-L | | 633 | 65.6 | 34.47 / 0.9280 | 30.35 / 0.8421 | 29.11 / 0.8053 | 28.32 / 0.8547 | 33.78 / 0.9458 |
| SwinIR-light | | 918 | 110.8 | 34.62 / 0.9289 | 30.54 / 0.8463 | 29.20 / 0.8082 | 28.66 / 0.8624 | 33.98 / 0.9478 |
| DLGSA-light | ×3 | 752 | 75.4 | **34.70** / 0.9295 | **30.58** / **0.8465** | **29.24** / 0.8089 | 28.83 / 0.8653 | 34.16 / 0.9483 |
| Omni-SR | | 780 | 88.4 | **34.70** / 0.9294 | 30.57 / 0.8469 | 29.28 / **0.8094** | **28.84** / **0.8656** | **34.22** / **0.9487** |
| **UPS** | | 832 | **72.4** | 34.66 / **0.9322** | 30.72 / **0.8489** | **29.31** / **0.8114** | 28.98 / 0.8685 | 34.53 / **0.9505** |
| SwinIR-S | ×3 | **503** | **47.9** | 34.38 / 0.9281 | 30.46 / 0.8448 | 29.15 / 0.8073 | 28.37 / 0.8572 | 33.77 / 0.9464 |
| **UPS-S** | ×3 | **459** | **40.4** | 34.53 / **0.9312** | 30.55 / 0.8463 | **29.24** / 0.8093 | 28.60 / 0.8614 | 34.12 / 0.9484 |
| Omni-SR+ | ×3 | 780 | 88.4 | 34.77 / 0.9304 | 30.70 / 0.8489 | 29.33 / 0.8111 | 29.12 / 0.8712 | 34.64 / 0.9507 |
| **UPS+** | ×3 | 832 | 72.4 | 34.78 / 0.9325 | 30.78 / 0.8492 | 29.36 / 0.8122 | 29.28 / 0.8728 | 34.84 / 0.9517 |
| IMDN | | 715 | 40.9 | 32.21 / 0.8948 | 28.58 / 0.7811 | 27.56 / 0.7353 | 26.04 / 0.7838 | 30.45 / 0.9075 |
| RFDN-L | | 643 | 37.4 | 32.28 / 0.8957 | 28.61 / 0.7818 | 27.58 / 0.7363 | 26.20 / 0.7883 | 30.61 / 0.9096 |
| SwinIR-light | | 930 | 63.6 | 32.44 / 0.8976 | 28.77 / 0.7858 | 27.69 / 0.7406 | 26.47 / 0.7980 | 30.92 / 0.9151 |
| DLGSA-light | ×4 | 761 | 42.5 | **32.54** / 0.8993 | **28.84** / **0.7871** | **27.73** / **0.7415** | **26.66** / **0.8033** | **31.13** / 0.9161 |
| Omni-SR | | 792 | 50.9 | 32.49 / 0.8988 | 28.78 / 0.7859 | 27.71 / **0.7415** | 26.64 / 0.8018 | 31.02 / 0.9151 |
| **UPS** | | 843 | **41.3** | 32.50 / **0.9024** | **28.90** / **0.7892** | **27.79** / **0.7435** | **26.83** / **0.8073** | **31.39** / **0.9194** |
| SwinIR-S | ×4 | **512** | **27.3** | 32.14 / 0.8955 | 28.67 / 0.7832 | 27.63 / 0.7382 | 26.22 / 0.7906 | 30.68 / 0.9111 |
| **UPS-S** | ×4 | **468** | **23.0** | 32.41 / **0.9008** | 28.80 / 0.7863 | **27.73** / 0.7414 | 26.58 / 0.7995 | **31.13** / **0.9163** |
| Omni-SR+ | ×4 | 792 | 50.9 | 32.57 / 0.8993 | 28.95 / 0.7898 | 27.81 / 0.7439 | 26.95 / 0.8105 | 31.50 / 0.9192 |
| **UPS+** | ×4 | 843 | 41.3 | 32.60 / 0.9029 | 28.97 / 0.7896 | 27.83 / 0.7446 | 27.10 / 0.8136 | 31.79 / 0.9223 |

# 5 Experiments

## 5.1 Settings

**Implementation Details.** Our UPS model is developed by PyTorch and incorporates several commonly used data augmentation techniques, including random cropping, vertical/horizontal flipping, and rotation. During training, we employ the Adam [35] optimization with cosine annealing [36], starting with an initial learning rate of $4e-4$. We set the batch size as 32 and the input image size as $64 \times 64$. Training is conducted for 600K iterations, utilizing four NVIDIA RTX 3090 GPUs.

**Scalable Model Size.** Generally, we train our UPS and UPS-S with different configurations. Our UPS model follows the setting of SwinIR-light [7], consisting of 4 D-RSTB blocks with 6 decoupled Swin Transformer layers (channel size: 60). Additionally, our UPS-S model is more lightweight with 4 compact D-RSTB blocks with varying numbers of decoupled Swin Transformer layers (6, 4, 4, 5) and a channel size of 48. Training various UPS models requires approximately 2-3 days.

**Benchmark Datasets.** Following previous studies [7, 28, 18], we utilize the DIV2K [37] image dataset for training. Subsequently, we conduct comprehensive evaluations on several widely-used SISR benchmarks, including Set5 [38], Set14 [39], BSD100 [11], Urban100 [40], and Manga109 [41]. Our quantitative comparison is based on PSNR and SSIM. Consistent with established research, we report the results specifically for the Y channel derived from the YCbCr color space.

## 5.2 Comparison with SOTA Methods

We perform extensive comparisons with a wide range of lightweight SISR models: MAFFSRN (ECCV20) [42], LAPAR-A (NeurIPS20) [18], LatticeNet (ECCV20) [19], RLFN (CVPRW22) [43], SwinIR-light [7], NGswin [28], SwinIR-NG [28], and DLGSA-l (ICCV23) [27]. More comprehensive comparisons with early SOTA lightweight models can be accessed in our supplementary material.

Table 2: Results of inference time (ms), FLOPs (G) and GPU memory usage (MB). The speed is tested on an NVIDIA GeForce RTX 2080Ti GPU with an input size of $256 \times 256$ under $\times 2$ lightweight SISR. FLOPs is calculated at an output resolution of 1280 x 720.

| Metrics | RFDN-L | LatticeNet | DLGSA-light | Omni-SR | SwinIR-light | UPS |
|---|---|---|---|---|---|---|
| Time (ms) ↓ | **13** | **18** | 225 | 112 | 175 | 119 |
| FLOPs (G) ↓ | **146** | 170 | 170 | 195 | 244 | **163** |
| Memory (GB) ↓ | **1577** | **1639** | 1800 | 1842 | 2051 | 1785 |

**Quantitative Comparison.** Tab.1 illustrates the quantitative evaluation. Our proposed UPS consistently outperforms existing methods across all benchmarks. Notably, UPS exceeds the second-best model by over 0.33dB on the Manga109 dataset[41] for the $\times 2$ setting. Additionally, our model shows significant improvements over SwinIR-light [7], achieving improvements of up to 0.5dB, 0.55dB, and 0.47dB on the Manga109 dataset [41] for the $\times 2$, $\times 3$, and $\times 4$ settings, respectively, while using fewer parameters. These results confirm the effectiveness of our decoupled optimization strategy. Significantly, when constrained by model complexities, our UPS demonstrates superior performance over SwinIR-light [7]. As shown in Tab.1, our approach achieves a 0.55dB increase in PSNR for $\times 2$ super-resolution on the Urban100 dataset[40], highlighting the advantages of our decoupled optimization in feature extraction and similarity modeling under compact parameter conditions.

Additionally, we evaluate the inference efficiency of various state-of-the-art (SOTA) lightweight single image super-resolution (SISR) models. As shown in Table 2, UPS reduces the overall inference cost by 33% in terms of FLOPs compared to our baseline model, SwinIR-light

Last, we have extensively explored the benefits of UPS for real-world SR and other frameworks, including HAT and DRCT, under both lightweight and parameter-intensive scenarios. Our experiments show that the proposed UPS consistently enhances efficiency and performance across all these settings (real-world SR, lightweight, and SISR tasks).

For real-world super-resolution (Real-world SR), as shown in Tab. 3 of the PDF file (also the table below), our proposed UPS-GAN outperforms other state-of-the-art GAN-based [44, 45] and even Diffusion-based methods (Reshift [46] and StableSR [47]) in terms of NIQE, NRQM, and PI metrics, achieving the best quantitative results (5.09/6.84/4.19). This confirms the effectiveness of UPS for real-world SR tasks.

**Qualitative Comparison.** Fig. 4 presents some visual examples. It is evident that UPS is capable of producing correct image textures with fewer super-resolved artifacts. Conversely, previous CNN-based and Transformer-based frameworks either fail to reconstruct clear image patterns or suffer from unpleasing artifacts. We provide more visual comparison in the supplementary material.

Table 3: Non-reference results of real-world SISR on RealSRSet [44].

| Metrics | BSRGAN | RealSR | ResShift† | StableSR† | SwinIR-GAN | UPS-GAN |
|---|---|---|---|---|---|---|
| NIQE ↓ | 5.66 | 5.83 | 8.37 | **5.24** | 5.49 | **5.09** |
| NRQM ↑ | 6.27 | 6.32 | 4.56 | 6.12 | **6.48** | **6.84** |
| PI ↓ | 4.75 | **4.40** | 7.03 | 4.66 | 4.72 | **4.19** |

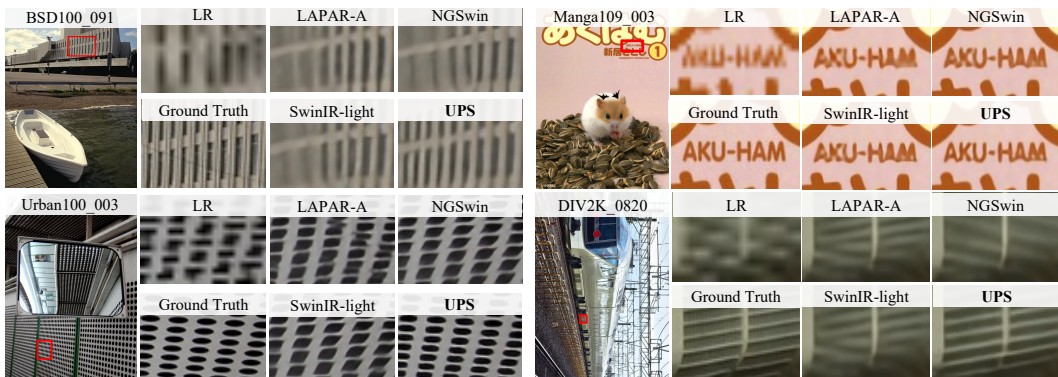

Figure 4: Qualitative comparison between LAPAR-A [18], SwinIR-light [7], NGSwin [28] and UPS (ours) on four popular benchmarks (BSD100 [11], Urban100 [40], Manga109 [41] and DIV2K [37].) under ×4 setting. Our predictions present more detailed textures and fewer artifacts.

## 5.3 Ablation Studies

To verify the effectiveness of our design, we conduct a series of comprehensive experiments. Note that we primarily adopt the UPS-S architecture for fast analysis. We generally report the quantitative results on the Set14 benchmark under the ×2 setting.

**Different Groups of Projection Space.** Unlike conventional attention-based frameworks, our approach introduces a unified projection space for similarity calculation. In this evaluation, we investigate the impact of incorporating additional groups of projection space. The quantitative results in Fig.5 (Left) shows that a higher number of block-wise projection groups reduce performance compared to our unified projection method. Additionally, layer-specific projection optimization exhibits inferior performance. Fig.5 (Right) confirms that our unified projection-sharing scheme outperforms models with multiple projection spaces. This experiment highlights the challenges of coupled optimization in image feature extraction and similarity modeling.

**Similarity Calculation.** Differing from the conventional similarity calculation paradigm (Matrix dot product + SoftMax), we incorporate the Cosine distance followed by a ReLU activation for similarity computation. Here, to assess the effectiveness of our choice, we train different models that utilize various combinations of distance metrics (Matrix dot and Cosine) and activation functions (SoftMax and ReLU). The results, as displayed in the left Fig. 6, indicate that our design (Cosine + ReLU) achieves the best performance among all the competing strategies. Additionally, we provide a corresponding visual comparison in the right Fig. 6. It shows that our design activates more non-local pixels, resulting in a larger valid receptive field and more precise reconstructed image details.

**Projection Matrix Dimension.** We investigate the impact of various projection dimensions for similarity calculation, ranging from 8 to 256 ($D = 8, 32, 64, 128, 256$). Tab. 4 indicates that performance initially improves as the projection dimension increases, but slightly drops after reaching extremely high dimensions (e.g., 256). We also observe that higher projection dimensions lead to increased computational costs on both learnable parameters and FLOPS.

Table 4: The effect of varying projection dimensions on similarity calculation.

| Dimension | 8 | 32 | 64 | 128 | 256 |
|---|---|---|---|---|---|
| PSNR (dB) | 33.83 | 33.86 | 33.90 | **34.00** | 33.96 |
| SSIM | 0.9206 | 0.9207 | 0.9207 | 0.9220 | **0.9221** |
| #Params | 452K | 452K | 452K | 453K | 454K |
| FLOPS | 107G | 113G | 119G | 124G | 159G |

| Proj. Group | G19 | G4 | G2 | G1 (Ours) |
|---|---|---|---|---|
| PSNR (dB) | 33.63 | 33.91 | 33.97 | **34.00** |
| SSIM | 0.9186 | 0.9205 | 0.9216 | **0.9220** |

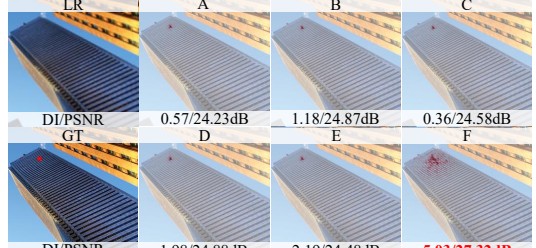

Figure 5: Analysis of several UPS-S models with different projection groups, including the layer-specific projection model (consists of 19 attention layers). (Left): PSNR/SSIMs are examined for quantitative comparison. (Right): Visual results on the Urban100 [39] benchmark under x2 setting.

| | Matrix dot | Cosine | SoftMax | ReLU | PSNR/SSIM |
|---|---|---|---|---|---|
| A | ✓ | | | | 33.60/0.9192 |
| B | ✓ | | ✓ | | 33.84/0.9202 |
| C | ✓ | | | ✓ | 33.41/0.9169 |
| D | | ✓ | | | 33.61/0.9208 |
| E | | ✓ | ✓ | | 33.73/0.9194 |
| F | | ✓ | | ✓ | **34.00/0.9220** |

Figure 6: Impact of different similarity calculation methods. The left Table shows the quantitative results of employing different similarity calculation methods on the Urban100 [39] ($\times 2$). The right figure gives a visual example to illustrate the SR results overlaid the LAM [33] maps of each model. The numbers beneath are the DI ($\uparrow$) [33] and PSNR ($\uparrow$) values.

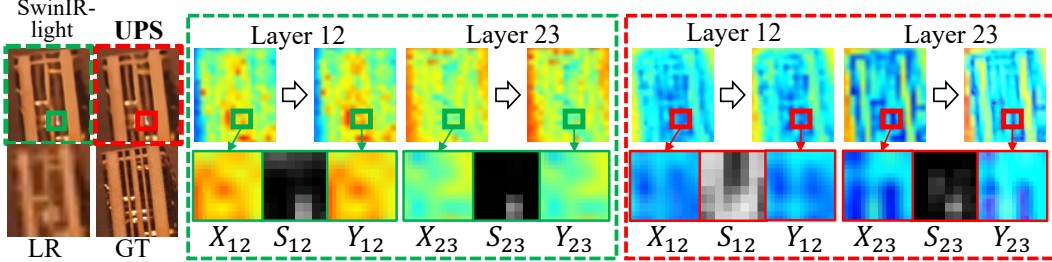

Figure 7: Visual comparison of layer-specific projection optimization and our proposed UPS scheme. UPS achieves better similarity calculation and yields better image structural restoration.

**Similarity Map Visualization.** To better understand the effect of our projection-sharing scheme, we visualize the deep image features of our baseline model (SwinIR-light) and UPS. For an LR image, Fig. 7 illustrates the updating of deep image features at layers $i = 12, 23$ in both the base and UPS models. The right arrows indicate the input-output data flow for each layer. Following that, we present a detailed visualization for Alg.1,2 in our paper[7]. Notably, the similarity maps $S_{12}, S_{23}$ produced by UPS (highlighted in the **red box**) are more effective in aggregating neighboring pixels, resulting in sharper final SR image.

## 5.4 Extension 1: Image Denoising and JPEG Image Deblocking

In this section, we explore the benefits of applying UPS for other image restoration tasks. While SwinIR (baseline model) requires millions of parameters for image denoising and deblocking, our lightweight UPS frameworks handle these common low-level restoration tasks more efficiently. As shown in Fig. 8, the results, indicate that UPS achieves performance comparable to the large baseline model (SwinIR) while requiring only $\frac{1}{13}$ of the model complexity on Denoising. Additionally, the compact baseline models exhibit inferior performance relative to our UPS. We describe the framework of these models in Sec. A.6 of our appendix.

---

[7]$X_i, Y_i$ denote input and output, and $S_i$ is the similarity matrix.

| Tasks | Metrics | SwinIR | SwinIR-C | UPS |
|---|---|---|---|---|
| Deblocking | PSNR | 29.86 | 29.63 | **29.98** |
| $q = 40$ | Param. | 11.50M | 3.89M | **3.49M** |
| Denoising | PSNR | **28.56** | 28.20 | 28.37 |
| $\sigma = 50$ | Param. | 11.50M | 0.959M | **0.873M** |

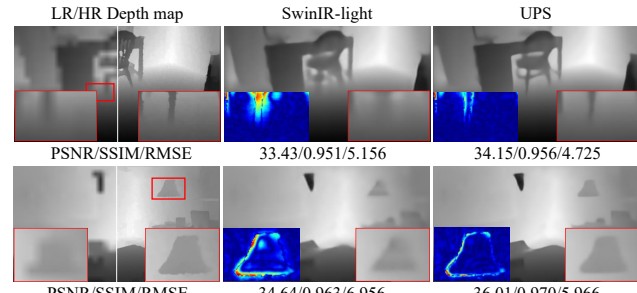

Figure 8: Extension on other image restoration problems. (Left) UPS attains comparable results compared with its larger baseline SwinIR and outperforms SwinIR-C with similar model sizes. (Right) A visual example of image deblocking.

| Settings | Metrics | SwinIR-light | UPS |
|---|---|---|---|
| | PSNR | 47.25 | **47.79** |
| ×4 | SSIM | 0.994 | **0.995** |
| | RMSE | 2.339 | **2.198** |
| | PSNR | 37.25 | **37.98** |
| ×16 | SSIM | 0.969 | **0.972** |
| | RMSE | 7.832 | **7.236** |

Figure 9: Generalization comparison between our baseline model and UPS. The quantitative results on NYU V2 [48] (×4, ×16) are displayed in the left table, while the right figure illustrates two visual examples. Additionally, normalized error maps are included in the left corner to facilitate comparison.

### 5.5 Extension 2: Depth Map Super-resolution

We also compare it with our baseline model (SwinIR-light) on the depth SR task. To do this, **without** training on any depth images, we directly test the two models on the NYU V2 [48] depth benchmark[8] under ×4 and ×16 settings. We use the PSNR, SSIM, and RMSE (the root-mean-square error) metrics for quantitative evaluation. The quantitative results are shown in the left Fig. 9. We can see UPS consistently outperforms its baseline model on all the objective metrics. For instance, UPS exhibits superior performance compared to SwinIR-light with a PSNR improvement of **0.54dB** (**0.73dB**) for ×4 (×16) configurations. The visual examples in the right Fig. 9 illustrate that UPS generates clearer structures, leading to higher accuracy when compared to our baseline model.

In addition to these two extensions, we also explore the improvement over SwinIR, DRCT, HAT for both lightweight and classic SISR to comprehensive analyze the potential capabilities of the proposed UPS. Please refer to the appendix sections.

## 6 Conclusion

In this paper, we propose a unified projection sharing (UPS) technique for lightweight SISR. A layer-invariant projection space is optimized for similarity modeling. Comprehensive experiments have demonstrated the effectiveness of the proposed decoupled learning algorithm. Notably, UPS achieves state-of-the-art performance on multiple SISR benchmarks. Moreover, UPS-S exhibits competitive results compared with leading approaches, while requiring fewer learnable parameters. Additionally, experiments indicate that our proposed UPS demonstrates superior data efficiency. Code will be made publicly available at `https://github.com/redrock303/UPS-NeurIPS2024`.

**Acknowledgments.** This work is partially supported by Shenzhen Science and Technology Program KQTD20210811090149095 and also the Pearl River Talent Recruitment Program 2019QN01X226. The work was supported in part by the Basic Research Project No. HZQB-KCZYZ-2021067 of Hetao Shenzhen-HK S&T Cooperation Zone, Guangdong Provincial Outstanding Youth Fund (No. 2023B1515020055), the National Key R&D Program of China with grant No. 2018YFB1800800, by Shenzhen Outstanding Talents Training Fund 202002, by Guangdong Research Projects No. 2017ZT07X152 and No. 2019CX01X104, by Key Area R&D Program of Guangdong Province (Grant No. 2018B030338001) by the Guangdong Provincial Key Laboratory of Future Networks of Intelligence (Grant No. 2022B1212010001), and by Shenzhen Key Laboratory of Big Data and Artificial

---

[8]NYU V2 consists of 449 testing depth images at a size of $480 \times 640$. We use bicubic interpolation to generate the low-resolution depth images.

Intelligence (Grant No. ZDSYS201707251409055). It is also partly supported by NSFC-61931024, NSFC-62172348, and Shenzhen Science and Technology Program No. JCYJ20220530143604010.

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

# A  Appendix / supplemental material

## A.1  Improvements over SOTA Swin-transformers

In our main paper, we have explore the improvements over SwinIR-light for lightweight SISR task. Here, we discuss the potential improvement and generalization capability of UPS for more Swin-transformers (such as DRCT [49] and HAT [21]) on both lightweight and classic SISR. Table. 5 shows our UPS is capable of enhancing several SOTA Swin-transformers with fewer parameters, FLOPs and inference latency.

Table 5: Quantitative comparison with SOTA **lightweight** models for ×2 SISR. All the models are trained on the DIV2K dataset for fair comparison. Inference time is tested at an input size of $256 \times 256$ on an NVIDIA GeForce RTX 2080Ti GPU.

| Lightwight | Params/FLOPs/Time (K / G / ms) | Set5 PSNR / SSIM | Set14 PSNR / SSIM | BSD100 PSNR / SSIM | Urban100 PSNR / SSIM | Manga109 PSNR / SSIM |
|---|---|---|---|---|---|---|
| SwinIR-light | 910 / 244 / 175 | 38.14 / 0.9611 | 33.86 / 0.9206 | 32.31 / 0.9012 | 32.76 / 0.9340 | 39.12 / 0.9783 |
| SwinIR-light-UPS | 843 / 163 / 119 | **38.26 / 0.9642** | **34.16 / 0.9232** | **32.42 / 0.9031** | **33.08 / 0.9373** | **39.62 / 0.9800** |
| DRCT-light | 1137 / 137 / 92 | 38.05 / 0.9632 | 33.76 / 0.9201 | 32.28 / 0.9012 | 32.48 / 0.9318 | 38.87 / 0.9783 |
| DRCT-light-UPS | 996 / 125 / 85 | **38.06 / 0.9634** | **33.89 / 0.9213** | **32.30 / 0.9013** | **32.59 / 0.9325** | **39.27 / 0.9786** |
| HAT-light | 813 / 102 / 153 | 38.02 / 0.9612 | 33.88 / 0.9203 | 32.28 / 0.9016 | 32.64 / 0.9330 | 38.82 / 0.9783 |
| HAT-light-UPS | 777 / 91 / 136 | **38.16 / 0.9636** | **34.16 / 0.9223** | **32.36 / 0.9022** | **32.92 / 0.9351** | **39.35 / 0.9791** |

Table 6: Quantitative comparison with SOTA **Classic** models for ×4 SISR. All the models are trained on the DF2K dataset for fair comparison. Inference time is tested at an input size of $256 \times 256$ on an NVIDIA GeForce RTX 2080Ti GPU.

| Classic | Param./FLOPs (G)/Time (Millions / G / ms) | Set5 PSNR / SSIM | Set14 PSNR / SSIM | BSD100 PSNR / SSIM | Urban100 PSNR / SSIM | Manga109 PSNR / SSIM |
|---|---|---|---|---|---|---|
| SwinIR | 11.9 / 584 / 683 | 32.92 / 0.9044 | 29.09 / 0.7950 | 27.92 / 0.7489 | 27.45 / 0.8254 | 32.03 / 0.9260 |
| SwinIR-UPS | 10.73 (-1.17) / 471 (-113) / 542 (-141) | **33.29 / 0.9116** | **29.51 / 0.8027** | **28.24 / 0.7595** | **28.03 / 0.8538** | **32.96 / 0.9332** |
| HAT | 20.77 / 728 / 1419 | 33.04 / 0.9056 | 29.23 / 0.7973 | 28.00 / 0.7517 | 27.97 / 0.8368 | 32.48 / 0.9292 |
| HAT-UPS | 17.25 (-3.52) / 633 (-95) / 1224 (-195) | **33.11 / 0.9098** | **29.29 / 0.7991** | **28.08 / 0.7548** | **28.61 / 0.8479** | **32.87 / 0.9319** |
| DRCT | 14.14 / 520 / 811 | 33.11 / 0.9064 | 29.35 / 0.7984 | 28.18 / 0.7532 | 28.06 / 0.8378 | 32.59 / 0.9304 |
| DRCT-UPS | 12.31(-1.83) / 482(-38)/ 669(-142) | **33.17 / 0.9088** | **29.38 / 0.7989** | **28.20 / 0.7536** | **28.32 / 0.8416** | **32.68 / 0.9331** |

## A.2  Data Efficiency

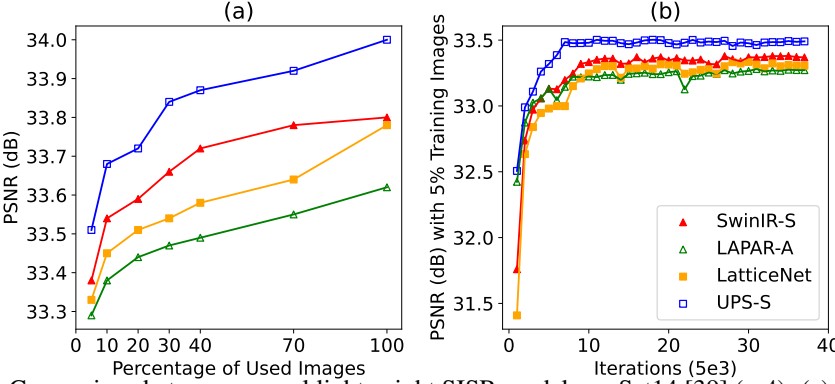

Figure 10: Comparison between several lightweight SISR models on Set14 [39] (×4). (a) Evaluation of using different numbers of training samples. (b) Validation performance of different models.

In this part, We assess the data efficiency of several models, including LAPAR-A, LatticeNet, our baseline model SwinIR-S [7] (a more lightweight model with identical framework configuration as our UPS-S), and our proposed UPS-S. In addition to training all models on the complete training set, we gradually adjust the percentage of used training data. As shown in Fig.10(a), UPS consistently outperforms other models regardless of training data size, demonstrating superior data efficiency. Fig.10(b) shows PSNR values during training iterations, with UPS-S converging faster and providing better early predictions. These results highlight the effectiveness of our UPS scheme.

## A.3    Robustness Optimization of UPS

Intuitively, given noisy input features, the error will be invertibly accumulated and affect the following projection space optimization.

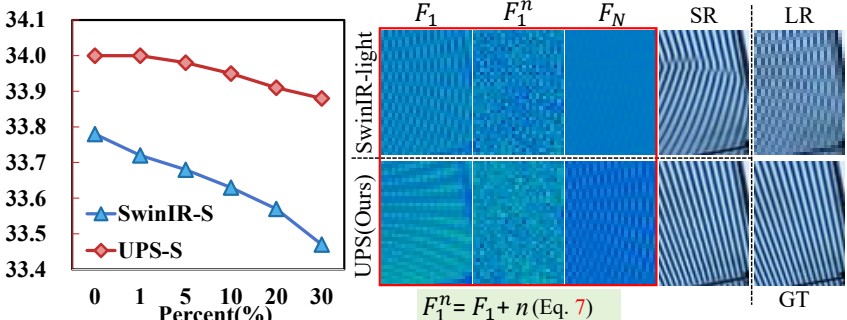

Figure 11: Comparison between SwinIR and our proposed UPS under noise optimization setup. We also report visual examples of the 'input feature $F_1$', 'input feature with noise (the noise std is set to 0.3) $F_1^n$', and 'output feature $F_N$' enhanced by methods and corresponding final predictions.

To better understand the effectiveness and robustness of our unified projection-sharing scheme, we compare SwinIR [7] and UPS under noisy input features to simulate the perturbation training. Firstly, we train both two models with perfect training images. Then, we add different levels of Gaussian white noise (zero mean and std values ranged from 0.01 to 0.3) on the input image feature $F_1$ (the input of the FEA module) for both training and evaluation:

$$F_1^n = F_1 + n, \tag{7}$$

where $n$ is the Gaussian white noise. The results are shown in Fig. 11. We observe that, with severe noise, the SwinIR-light fails to restore high-frequency details in the output feature $F_N$ produced by the last STL layer, thus it produces incorrect image structures. In contrast, UPS is able to recover high-frequency signals from the noisy input feature and accordingly reconstruct accurate image details. This experiment suggests our proposed UPS is more optimization robust and effective with noisy input features.

Table 7: Robustness comparison of SwinIR-light [7], NGSwin [28] and our proposed UPS. While being trained on clean DIV2K [37] training samples, we directly evaluate their generalization ability on the degraded Set14 benchmark under the x4 setting. We report the PSNR (dB)/SSIM values for quantitative evaluation.

| Degradation | None | Compression | Blur | Noise |
|---|---|---|---|---|
| SwinIR-light | 28.77/0.7858 | 28.45/0.7783 | 28.69/0.7829 | 27.93/0.7562 |
| NGSwin | 28.83/0.7870 | 28.46/0.7783 | 28.65/0.7819 | 27.93/0.7532 |
| UPS | **28.90/0.7892** | **28.73/0.7838** | **28.87/0.7864** | **28.29/0.7616** |

## A.4    Robustness on Degraded Data

The co-adaptation issue [8, 9] reveals that deep-learning models may fit the training samples well but exhibit poor generalization for unseen data, especially for out-of-domain samples. Here, we aim to evaluate the robustness of SOTA SISR models (including SwinIR-light and NGSwin) and our proposed UPS. To do this, we apply different data degradations (i.e., JPEG compression, Gaussian blur, and Gaussian White noise) and obtain degraded samples. Without training on these degradations, we directly evaluate the performance of these lightweight SISR models. The results presented in Tab. 7 illustrate our proposed UPS is more robust on unseen data. Notably, UPS outperforms the competing models up to 0.36dB on the noise testing data. Moreover, Fig. 12 shows both SwinIR-light and NGSwin restore incorrect and blur image structures due to their poor generalization (not robust) for unseen degradations. In contrast, UPS produces more accurate results with clearer image contents.

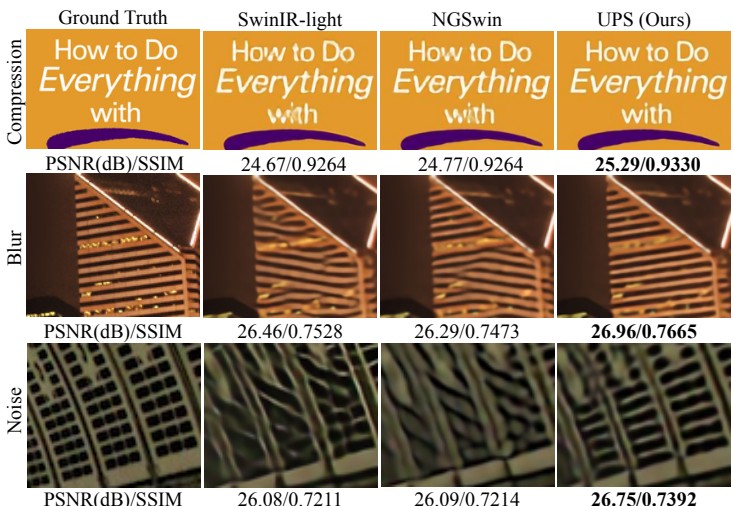

Figure 12: Qualitative robustness comparison of different lightweight SISR models on out-of-domain degraded inputs under x4 setting. The first sample is from Set14 and the last two samples are from the Urban100 benchmark.

Table 8: More ablation studies (PSNR/SSIM). **A**: The impacts of ReLU and Softmax activations in Eq.6. **B**. Quantitative comparison between two advanced optimization schemes (Dropout in RDSR CVPR 2022 and progressive training in DRCT ARXIV 2024.) and UPS.

| Analysis | **A**. Activation | | | **B**. Optimization schemes vs. UPS | | | |
|---|---|---|---|---|---|---|---|
| | SwinIR-light | UPS (Softmax) | UPS (ReLU) | SwinIR-light | SwinIR + Dropout | SwinIR + Pro. Train | UPS |
| Urban100 (×4) | 26.47 / 0.7980 | 26.79 / 0.8069 | **26.83 / 0.8073** | 26.47 / 0.7980 | 26.52 / 0.7988 | 26.56 / 0.7986 | **26.83 / 0.8073** |
| Improve. | - | **+0.32 / +0.0089** | **+0.36 / +0.0093** | - | +0.05 / +0.0008 | +0.09 / +0.0006 | **+0.36 / +0.0093** |
| Param. (K) | 930 | **843 (-87)** | **843 (-87)** | 930 | 930 | 930 | **843 (-87)** |

## A.5 More Ablated Studies

**Impact of ReLU and Softmax.** We conduct the experiment to demonstrate the impact of different activations. The results in Tab.8(**A**) suggests the used ReLU performs better than the softmax activation. As we can see, the main improvement comes from our UPS design instead of the ReLU activation. The performance gap between the two different activation choices is only 0.04dB, which represents 11% of the total improvement of 0.36dB. In other words, the 89% improvements come from the UPS design. We will include this ablation analysis in our revised paper.

**UPS vs. Other Optimization Schemes.** We investigate two existing training strategies for the SISR task. RDSR [9] incorporates dropout techniques to achieve better testing results, while DRCT [49] employs a progressive training scheme that involves multi-stage training to enhance final performance. Here, we compare UPS with RDSR and progressive training schemes in DRCT. To do this, we re-train SwinIR-light using the above two training strategies. As shown in Tab.8(**B**), UPS delivers superior results compared to both of these optimization methods. Nevertheless, we hope our exploration will inspire future research to develop more effective algorithms to better address this challenge.

**The Identity Mapping of $X_i$ and $V_i$.** For the lightweight scenario, we aim to further reduce the computational cost and model size. Thus, we explore cutting off the linear mapping between $X_i$ and $V_i$, and our early experimental analysis (presented in Tab. 9) suggests such a design will not lead to a performance drop. We will add this discussion to our revised paper.

## A.6 Framework Details for Image Denoising and Deblocking

In Section 5.4, we delve into the advantages of UPS in tasks like image denoising and JPEG removal, conducting a comparative analysis among various baseline models and UPS. For both tasks, the SwinIR model adheres to the default framework settings outlined in the original paper, featuring 6 RSTB blocks with a channel size of 180. SwinIR-C and UPS follow similar framework configurations:

Table 9: **C**. $V$ projection indicates the linear projection for transforming the input $X_i$ into $V_i$.

| Results&Param. | w/ V proj. | w/o V proj. (Default) |
|---|---|---|
| Urban100 ($\times 4$) | 26.80 / 0.8071 | **26.83 / 0.8073** |
| Set14 ($\times 4$) | **28.91 / 0.7892** | 28.90 / **0.7892** |
| Param. (K) | 895 | **843** |

8 RSTB/D-RSTB blocks with a channel dimension of 90 for image deblocking, and 6 RSTB/D-RSTB blocks with a channel size of 60 for image denoising.

## A.7 Limitation

While UPS exhibits SOTA results in lightweight SISR, we have not investigated its potential benefits for large (UPS-based) models. Exploring larger UPS-based models is an interesting future work. On the other hand, we will explore more applications for a wide range of low-level tasks, such as real-world image restoration.

