# OpenReview forum: "UPS: Unified Projection Sharing for Lightweight Single-Image Super-resolution and Beyond"
_NeurIPS.cc/2024/Conference — NeurIPS 2024 poster_

### Official Review · Reviewer_HE6k · 2024-07-09

**Soundness:** 3
**Presentation:** 3
**Contribution:** 2
**Rating:** 4
**Confidence:** 5

**Summary:**

This paper introduces Unified Projection Sharing (UPS), a novel algorithm for lightweight single-image super-resolution (SISR) that decouples feature extraction from similarity modeling by employing a unified projection space. This approach achieves state-of-the-art performance across various benchmarks while demonstrating robustness for unseen data and promising results for additional image restoration tasks, all with a computationally efficient model.

**Strengths:**

1. This paper has a clear writing approach and brings new insights.
2. The UPS algorithm demonstrates superior performance compared to existing lightweight SISR methods across multiple benchmarks, showcasing its effectiveness.
3. The method shows promise for extension to other image restoration tasks beyond SISR, suggesting its potential for broader application in image processing.

**Weaknesses:**

1. The author did not conduct experiments outside of the lightweight model [1][2], and from an optimization perspective, the proposed method should be possible to improve performance in various situations. The authors need to be able to justify themselves.

2. Lack of actual latency comparison. Due to some differences in network structure, the params and FLOPs may not accurately reflect the efficiency of the model.

3. Lack of comparison with the SOTA model, e.g., [3]. outperforms the UPS almost on every benchmark.

Ref:

1. Chen, Xiangyu et al. “Activating More Pixels in Image Super-Resolution Transformer.” 2023 IEEE/CVF Conference on Computer Vision and Pattern Recognition (CVPR) (2022): 22367-22377.

2. Hsu, Chih-Chung et al. “DRCT: Saving Image Super-resolution away from Information Bottleneck.” ArXiv abs/2404.00722 (2024): n. pag.

3. Wang, Hang et al. “Omni Aggregation Networks for Lightweight Image Super-Resolution.” 2023 IEEE/CVF Conference on Computer Vision and Pattern Recognition (CVPR) (2023): 22378-22387.

**Questions:**

See above.

**Limitations:**

Yes.

---

> ### Author Rebuttal · Authors · 2024-08-07
>
> **Q1. Conduct experiments outside of the lightweight model...**
>
> Thanks for your thoughtful comments. As you appreciated the potential generalization of UPS on more image restoration tasks, e.g., JPEG compression removal and image de-noising tasks, we explore the potential application of UPS for common SISR and real-world SR tasks and more baseline frameworks (e.g., HAT/HAT-light, DRCT/DRCT-light) and report the results in Tab. 2 (a) and Tab. 3/4 of our global PDF response.
>
>
> For both common and lightweight single image super-resolution (SISR), we incorporate UPS into other state-of-the-art SISR models, including SwinIR, HAT/HAT-light [4], and DRCT/DRCT-light [5], based on your suggestion. As shown in Tables 3 and 4 of our response file, we observe that UPS consistently improves the performance and inference efficiency of these models (HAT-UPS, DRCT-UPS, SwinIR-UPS, and their lightweight versions HAT-light-UPS, DRCT-light-UPS, and SwinIR-light-UPS). We will include these insightful experiments in our revised paper. We believe your insightful suggestion and our exploration enhance the wide-ranging value of UPS.
>
> Additionally, for real-world super-resolution (SR), we follow our baseline model, SwinIR-GAN (trained using BSRGAN degradation), and train UPS-GAN. We evaluate all real-world SR models using the same benchmarks (RealSRSet) and evaluation metrics as SwinIR-GAN. As shown in Table 2a of the PDF file (also the Tab below), our proposed UPS-GAN outperforms other state-of-the-art GAN-based and even Diffusion-based methods (Reshift NeurIPS 2023[1] and StableSR IJCV 2024[2]) in terms of NIQE, NRQM, and PI metrics, achieving the best quantitative results (5.09/6.84/4.19). This confirms the effectiveness of UPS for real-world SR tasks.
>
> | Metrics | BSRGAN | RealSR |  ResShift |  StableSR  | SwinIR-GAN | UPS-GAN |
> |---------|--------|--------|-----------|-----------|------------|---------|
> | NIEQ ↓  | 5.66   | 5.83   | 8.37      | 5.24  | 5.49       | **5.09** |
> | NRQM ↑  | 6.27   | 6.32   | 4.56      | 6.12      | 6.48   | **6.84** |
> | PI ↓    | 4.75   | 4.40| 7.03     | 4.66      | 4.72       | **4.19** |
>
>
>
> **Q2. Latency comparison**
>
> Please see the first answer of the global response. We will provide all these discussions in our revised paper.
>
>
> **Q3. Comparison with the SOTA model, e.g., Omni-SR**
>
> As discussed in our paper (Line 175), we use the widely used DIV2K for training. When compared with Omni-SR [6] (trained on the same DIV2K), UPS surpasses it in most cases.
>
> To make a fair comparison with Omni-SR+ (trained on DF2K), we re-trained UPS+ from scratch on DF2K. We can see that UPS+ generally attains superior performance than Omni-SR+ in the below table. For instance, UPS+ achieves 0.27dB/0.20dB/0.29dB improvements than Omni-SR+ on Manga109 x{2,3,4}.
>
> | Method    | Scale|  Set5          | Set14         | BSD100        | Urban100       | Manga109       |
> |-----------|---------------|---------------|---------------|---------------|----------------|----------------|
> | Omni-SR  | ×2 | 38.22 / 0.9613 | 33.98 / 0.9210 | 32.36 / 0.9020 | 33.05 / 0.9363 | 39.28 / 0.9784 |
> | UPS      | ×2 | **38.26 / 0.9642** | **34.16 / 0.9232** | **32.42 / 0.9031** | **33.08 / 0.9373** | **39.62 / 0.9800** |
> | | | | | | |
> | Omni-SR+ | ×2 | 38.29 / 0.9617 | 34.27 / 0.9238 | 32.41 / 0.9026 | 33.30 / 0.9386 | 39.53 / 0.9792 |
> | UPS+    | ×2  | **38.31 / 0.9643** | **34.37 / 0.9247** | **32.43 / 0.9032** | **33.34 / 0.9388** | **39.80 / 0.9802** |
> | | | | | | |
> | Omni-SR    | ×3     | **34.70** / 0.9294   | 30.57 / 0.8469    | 29.28 / 0.8094 | 28.84 / 0.8656   | 34.22 / 0.9487       |
> | UPS        | ×3    | 34.66 / **0.9322** | **30.72** / **0.8489**    | **29.31** / **0.8114**| **28.98** / **0.8685**| **34.53** / **0.9505**|
> | | | | | | |
> | Omni-SR+ | ×3 |34.77 / 0.9304 | 30.70 / 0.8489 | 29.33 / 0.8111 |  29.12 / 0.8712 | 34.64 / 0.9507 |
> | UPS+       | ×3     | **34.78 / 0.9325**   | **30.78 / 0.8492**    | **29.36 / 0.8122**     | **29.28 / 0.8728**       | **34.84** / **0.9517**       |
> |            |       |                 |           |                  |                   |                    |                      |                      |
> | Omni-SR    | ×4    | 32.49 / 0.8988 | 28.78 / 0.7859 | 27.71 / 0.7415 | 26.64 / 0.8018 | 31.02 / 0.9151      |
> | UPS        | ×4    | **32.50** / **0.9024**| **28.90 / 0.7892**    | **27.79** / **0.7435**| **26.83** / **0.8073**| **31.39** / **0.9194**|
> | | | | | | |
> | Omni-SR+ | ×4 |32.57 / 0.8993 |  28.95 / 0.7898 |  27.81 / 0.7439 | 26.95 / 0.8105 | 31.50 / 0.9192|
> | UPS+       | ×4    | **32.60 / 0.9029**   | **28.97 / 0.7896**    | **27.83 / 0.7446**     | **27.10 / 0.8136**       | **31.79 / 0.9223**       |
>
> More results can be found in Tab. 1 of our response PDF. We will include this comparison in Tab. 1 of our revised paper and highlight the two different training setups to ensure a clear and fair comparison.
>
> ## References
> [1] Yue, et al. ""Resshift: Efficient diffusion model for image super-resolution by residual shifting." NeurIPS, 2023"
>
> [2] Wang et al. "Exploiting Diffusion Prior for Real-World Image Super-Resolution." IJCV, 2024.
>
> [3] Liang, et al. "Swinir: Image restoration using swin transformer." CVPR, 2021.
>
> [4] Chen, et al. “Activating More Pixels in Image Super-Resolution Transformer.” CVPR, 2022.
>
> [5] Hsu, et al. “DRCT: Saving Image Super-resolution away from Information Bottleneck.” arxiv, 2024.
>
> [6] Wang, et al. “Omni Aggregation Networks for Lightweight Image Super-Resolution.” CVPR, 2023.

---

> ### Author Response · Authors · 2024-08-14
>
> Dear Reviewer HE6k:
> We have made every effort to address all concerns and provide comprehensive evidence.
>
> For Q1, we explored real-world SR and large models for SISR, and also integrated UPS into DRCT/DRCT-light and HAT/HAT-light to boost their performance.
>
> For Q2, as acknowledged by all other reviewers, we provided inference efficiency for further comparison.
>
> For Q3, we conducted a detailed comparison with Omni-SR and Omni-SR+, showing that UPS/UPS+ outperforms them under a fair setup (using DIV2K/DF2K for training).
>
>
> Could you please let us know if our rebuttals and further responses have answered all your questions? We greatly appreciate it.

---

### Official Review · Reviewer_RnSV · 2024-07-09

**Soundness:** 3
**Presentation:** 3
**Contribution:** 2
**Rating:** 4
**Confidence:** 5

**Summary:**

This work introduces a novel unified projection sharing algorithm that decouples feature extraction and similarity modeling. A unified projection space defined by a learnable projection matrix is created for similarity computation across all self-concerned layers. Extensive experiments demonstrate that the proposed UPS achieves state-of-the-art performance compared to leading lightweight SISR methods.

**Strengths:**

1. This paper is clearly written and well organized.

2. The sharing of unified projections can effectively reduce computation and enable performance improvements.

3. Experiments show the promising performance of the proposed method.

**Weaknesses:**

1. The idea of shared projection is somewhat similar to attention sharing [1,2,3]. Discussion and analysis with these highly relevant studies is needed.

2. The needs in Table 1 include computational quantities for the different models. It is unfair to compare with CNN-based methods only the parameters, which are much less computationally intensive.

3. Inference efficiency must be included in the comparison, including CNN and Transformer based methods. Paramas and FLOPs do not fully reflect the on-device running speed of the model.

4. Lack of experimental evaluation in more complex real-world scenarios.

5. Similarity calculation methods have been discussed in past studies [4, 5, 6].

> 1.  ShareFormer: Share Attention for Efficient Image Restoration. arxiv 2023.

> 2. Skip-Attention: Improving Vision Transformers by Paying Less Attention. arxiv 2023.

> 3. You Only Need Less Attention at Each Stage in Vision Transformers. CVPR 2024.

> 4. EfficientViT: Multi-Scale Linear Attention for High-Resolution Dense Prediction. ICCV 2023.

> 5. Swin transformer v2: scaling up capacity and resolution. CVPR 2022.

> 6. Swin2SR: SwinV2 Transformer for Compressed Image Super-Resolution and Restoration. ECCVW 2022.

**Questions:**

Please address the issues raised in the Weaknesses.

**Limitations:**

Limitations were discussed.

---

> ### Author Rebuttal · Authors · 2024-08-07
>
> **Q1. Discussion and analysis with attention sharing...**
>
> Thanks for your suggestion. ShareFormer [1] presents a local similarity map-sharing scheme between neighboring attention layers for lower latency. Thus, ShareFormer shares a static similarity map for neighboring attention layers while UPS calculates dynamic similarity maps with layer-refined features in a shared projection space.
>
> Skip-Attention cuts off some intermediate attention layers to improve efficiency and performance for high-level tasks. LaViT [2] proposes a residual-based attention downsampling that fuses the initial calculated attention scores to guide the aggregation of the following layers, resulting in faster efficiency and improved classification accuracy.
>
> Therefore, Skip-Attention [3] and LaViT follow the existing coupled optimization scheme (reduce some attention calculations), and UPS proposes a decoupled learning strategy to enhance performance. We will cite the insightful studies and add this discussion to our revised paper.
>
> **Q2. Report the computational quantities of different models**
>
> Thanks. We agree with you on this point and we will include the FLOPs (like Fig. 1(c) of our paper) of different CNNs and Transformers for a comprehensive comparison. Please also refer to the FLOPs results in Tab. 1, 2 (b), 3 and Table 4 of our response PDF file.
>
> **Q3. Inference efficiency of different models**
>
> Please see the first answer of the global response and more inference comparison in the Tab. 1. 2(b), 3, 4 in our response PDF file. We will provide all these discussions in our revised paper.
>
> **Q4. Experimental evaluation real-world SR.**
>
>    Thanks for this good comments. Following SwinIR-GAN, our baseline model, we train UPS-GAN with the same configuration using the widely used BSRGAN degradation. We compare UPS-GAN with SwinIR-GAN, BSRGAN, RealSR, Reshift, and StableSR for real-SR task. We evaluate all real-world SR models using the same benchmarks (RealSRSet) and assessment metrics as SwinIR-GAN. As shown in Table 2a of the PDF file (also the Tab below), our proposed UPS-GAN outperforms other state-of-the-art GAN-based and even Diffusion-based methods (Reshift NeurIPS 2023[7] and StableSR IJCV 2024[8]) in terms of NIQE, NRQM, and PI metrics, achieving the best quantitative results (5.09/6.84/4.19). This confirms the effectiveness of UPS for real-world SR tasks.
>
> | Metrics | BSRGAN | RealSR |  ResShift  |  StableSR  | SwinIR-GAN | UPS-GAN |
> |---------|--------|--------|-----------|-----------|------------|---------|
> | NIEQ ↓  | 5.66   | 5.83   | 8.37      | 5.24  | 5.49       | **5.09** |
> | NRQM ↑  | 6.27   | 6.32   | 4.56      | 6.12      | 6.48   | **6.84** |
> | PI ↓    | 4.75   | 4.40| 7.03     | 4.66      | 4.72       | **4.19** |
>
> **Q5. The adopted similarity calculation method in UPS**
>
> As you pointed out, apart from the ReLUFormer discussed in our method section, we will cite and discuss the related works in our revised paper.
>
> ### References
>
> [1] ShareFormer: Share Attention for Efficient Image Restoration. arxiv 2023.
>
> [2] You Only Need Less Attention at Each Stage in Vision Transformers. CVPR 2024.
>
> [3] Skip-Attention: Improving Vision Transformers by Paying Less Attention. arxiv 2023.
>
> [4] EfficientViT: Multi-Scale Linear Attention for High-Resolution Dense Prediction. ICCV 2023.
>
> [5] Swin transformer v2: scaling up capacity and resolution. CVPR 2022.
>
> [6] Swin2SR: SwinV2 Transformer for Compressed Image Super-Resolution and Restoration. ECCVW 2022.
>
> [7] Yue, et al. ""Resshift: Efficient diffusion model for image super-resolution by residual shifting." NeurIPS, 2023"
>
> [8] Wang et al. "Exploiting Diffusion Prior for Real-World Image Super-Resolution." IJCV, 2024.

---

> > ### Comment · Reviewer_RnSV · 2024-08-13
> > **Response to author's rebuttal**
> >
> > Thank you for your response. Concerns about FLOPs and running efficiency were addressed. However, I still have some issues regarding the evaluation of real-world SR. NIQE, NRQM, and PI are not commonly used metrics for evaluating real-world SR, which is typically measured using PSNR, SSIM, LPIPS, CLIPIQA, and MUSIQ. Additionally, the response does not discuss the differences and uniqueness of the similarity calculation strategy from previous work. Therefore, I am inclined to keep the original rating.

---

> > > ### Author Response · Authors · 2024-08-14
> > > **Dear Reviewer RnSV**
> > >
> > > We have made every effort to address all concerns and provide as much evidence as possible. Could you please let us know if our rebuttals and further responses have answered all your questions? We greatly appreciate it.

---

> ### Author Response · Authors · 2024-08-13
> **Response for the further problems**
>
> Thank you for your additional comments. We're pleased that our previous response addressed some of your concerns. Here are our further responses to the issues you've raised:
>
> 1. As acknowledged by you and the other reviewers, UPS is a novel and effective lightweight SISR algorithm. Nonetheless, we explored the potential benefits of applying UPS to this new task. We ensured a fair evaluation by strictly following the same baseline (SwinIR, BSRGAN) and using consistent evaluation metrics, such as NIQE, NRQM, and PI, for all competing methods on the RealSRSet [1] benchmark.
>
> 2. In response to Q1, we have thoroughly examined the contributions of the suggested works and outlined the key differences between them and UPS. To summarize, UPS operates independently from these transformers. For instance, while ShareFormer generates a static similarity map for neighboring layers, UPS conducts dynamic similarity calculations within a shared projection space. Furthermore, unlike these three works, which adhere to coupled layer-specific similarity and feature extraction optimization, UPS introduces a decoupled approach for these aspects, specifically designed for lightweight SISR.
>
> 3. Due to the absence of ground truth data in the RealSRSet [1] benchmark, we cannot calculate reference-based metrics like PSNR, SSIM, and LPIPS. Instead, we present results using CLIPIQA and MUSIQ, which are widely accepted in diffusion-based models. Although UPS-GAN follows the GAN-based framework of our baseline models (SwinIR-GAN), it consistently achieves top-1 and top-2 performance when compared to other leading methods, as shown in the Table below.
>
> | Metrics | BSRGAN | RealSR |  ResShift$\dagger$ |  StableSR$\dagger$ | SwinIR-GAN | UPS-GAN |
> |---------|--------|--------|-----------|-----------|------------|---------|
> | CLIPIAQ↑  | 0.6321   | 0.6051   | 0.5834      | 0.5025  | 0.5996       | **0.6577** |
> | MUSIQ ↑  | **65.85**  | 62.59   | 54.29      | 60.32      | 63.45   | 64.79 |
>
> 1. Zhang et al. "Designing a Practical Degradation Model for Deep Blind
> Image Super-Resolution." ICCV, 2021.

---

### Official Review · Reviewer_Pasd · 2024-07-11

**Soundness:** 3
**Presentation:** 3
**Contribution:** 3
**Rating:** 7
**Confidence:** 4

**Summary:**

The paper proposes an effective lightweight decoupled SISR algorithm that simultaneously performs layer-specific optimization for deep feature extraction and similarity modeling. Specifically, the proposed method casts the deep feature extraction as per-layer optimization, while the similarity modeling is achieved by shared projection space. The proposed method is able to be extended to many restoration tasks, such as denoising, and JPEG image deb locking.

**Strengths:**

1. The paper is well-written.
2. The authors bring a novel perspective for lightweight SISR that the deep feature extraction and similarity modeling can be decoupled.

**Weaknesses:**

1. Line 49: "simultaneously" -> "simultaneous".
2. Figure 2: the placement of S1 and Si in Fig. 2(c) is incorrect.
3. Eq. (5): why do you set Vi directly equal to Xi instead of projecting Xi to Vi as done in Swin Transformer?
4. Eq. (6): what does $Q_{i}^D$ denote in this equation? What does D represent?
5. It's recommended to report the quantitative results on the DIV2K dataset.

**Questions:**

See the Weaknesses.

**Limitations:**

If the authors can address my concerns, I am ready to change my recommendation based on the comments.

---

> ### Author Rebuttal · Authors · 2024-08-07
>
> We sincerely thank you for your appreciation of the novelty and effectiveness of our UPS for lightweight SISR.
>
> **Q1. A typo: Line 49...**
>
> We appreciate your kind comment. We will carefully revise our paper to fix the typo.
>
> **Q2. Figure 2: the misplacement of $S_1$ and $S_i$**
>
> Thanks, we will correct the placement of the $S_1$ and $S_i$ accordingly.
>
> **Q3. The indetity mapping of $X_i$ and $V_i$**
>
> For the lightweight scenario, we aim to further reduce the computational cost and model size. Thus, we explore cutting off the linear mapping between $X_i$ and $V_i$, and our early experimental analysis (presented in Tab. 5-C of our PDF file) suggests such a design will not lead to a performance drop. We will add this discussion to our revised paper.
>
> | Settings           |  w/ V proj.    |     w/o V proj (Default)        |
> | :----              |    :----                  |    :----             |
> | Urban100 (×4)      |         26.80/0.8071      |        26.83/0.8073 |
> | Set14 (×4)         |         28.91/0.7892      |        28.90/0.7892 |
> | Parameters (K)/FLOPs (G)         |             895/179           |           843/163       |
>
> **Q4. The denotation of $D$ in $Q_i^D$ in Eq. 6**
>
> $D$ means the projection dimension in our UPS. We are sorry for this typo. The correct Eq.6 should be:
>    $$
>    ReLU(Consine(Q_i,Q_i^T) + B_i),
>    $$
> where $T$ means the transpose operation for matrix multiplication. We will double-check all these equations to avoid any difficulties in understanding our work.
>
> **Q5. Quantitative results on the DIV2K dataset...**
>
> In this work, we adopt the DIV2K for training our models as indicated in Line 175 "we utilize the DIV2K image dataset for training". We will highlight this important training setting in our revised paper.

---

> > ### Comment · Reviewer_Pasd · 2024-08-08
> > **Rating**
> >
> > All concerns have been addressed here, so I‘d like to accept this paper (rating: 7).

---

> ### Author Response · Authors · 2024-08-08
> **Thanks for your comment**
>
> Thank you for your kind comments and appreciation of our work. We will do our best to improve the final version of our paper based on your suggestions.

---

### Official Review · Reviewer_fbd3 · 2024-07-12

**Soundness:** 2
**Presentation:** 2
**Contribution:** 2
**Rating:** 5
**Confidence:** 5

**Summary:**

This paper presents a novel algorithm named UPS designed to enhance the performance of Transformer-based frameworks in single-image super-resolution (SISR), particularly under lightweight scenarios. The authors identify the challenge posed by the simultaneous layer-specific optimization required for deep image feature extraction and similarity modeling in existing methods. To address this, UPS decouples these tasks by establishing a unified projection space via a learnable projection matrix. The proposed UPS method demonstrates state-of-the-art performance. Additionally, UPS shows good performance on broader image restoration applications.

**Strengths:**

1. This paper presents a simple yet effective lightweight method called UPS. Using this method, the training difficulty of the model is reduced, and the model effect is improved on the premise of reducing the number of parameters and FLOPS.
2. Sufficient quantitative experimental results are given in this paper.

**Weaknesses:**

1. “Fig. a.(1-3) below shows over 0.95% (0.99%, 0.95%, 0.96%) for ×{2, 3, 4}) (projection layer) pairs get over 0.9 scores (ranging from 0 to 1) ". If there are no errors in this paragraph, the similarity between layers is very low.
2. The contribution point in this paper is the Unified Projection Sharing. However, in the method, the activation function is also modified. I wonder how much of a performance boost I would get if I left the activation function unchanged and just used consistent space sharing. Please give the analytical or experimental results.
3. As a lightweight method, it would be better to compare inference time.

**Questions:**

In this paper, it is mentioned that each layer of the model carries out image feature extraction and similarity modeling at the same time, which is difficult to train and will affect the performance of the model. Then, can the model performance be improved by improving the training strategy and modifying the loss function? In theory, if you can train the model properly, will you get better performance than UPS?

**Limitations:**

The paper has discussed methodological limitations.

---

> ### Author Rebuttal · Authors · 2024-08-07
>
> **Q1. In line 30, "Fig. 1a.(1-3) below shows over 0.95% (0.99%, 0.95%, 0.96%) ..."**
>
> Thanks a lot for pointing out this typo, actually it's a writing error. It should be "Fig. 1a.(1-3) below shows over **0.95 (0.99, 0.95, 0.96)** ...". We will revise our manuscript to avoid misunderstanding.
>
> **Q2. Experimental results when the activation function is consistent with SwinIR-light**
>
> That is quite a profound question. We have provided the ablated analysis (Tab. 5-A) in the global response PDF file. Also, we report the specific quantitative results in the table below. As we can see, the main improvement comes from our UPS design instead of the ReLU activation. The performance gap between the two different activation choices is only 0.04dB, which represents 11% of the total improvement of 0.36dB. In other words, the 89% improvements come from the UPS design. We will include this ablation analysis in our revised paper.
>
> | Activation  Function    | SwinIR-light (base) | UPS (Softmax) |  UPS (ReLU, Default) |
> | :----        |    :----    |    :---- |   :----  |
> | Urban100 (×4)     |         26.47/0.7980      |         26.79 (+0.32)/ 0.8069  (+0.0089)           |       26.83 (+0.36)/0.8073 (+0.0093)   |
> | Parameters (K)           |             930                 |           843          |        843                                       |
>
> **Q3. To compare the inference time**
>
> Please see the first answer of the global response. We will provide all these discussions in our revised paper.
>
>
> **Q4. More effective optimization strategy may also enhance the performance**
>
> We agree with you on the point that an advanced training strategy may also boost the performance of entangled SISR models. As praised by you and other Reviewers, UPS provides a simple yet effective optimization scheme:  without any special training strategies, e.g., taking additional training costs or carefully model finetuning, while yielding improved results.
>
>    We investigate two existing training strategies for the SISR task. RDSR [1] incorporates dropout techniques to achieve better testing results, while DRCT [2] employs a progressive training scheme that involves multi-stage training to enhance final performance. Here, we compare UPS with RDSR [1] and progressive training schemes in DRCT [2]. To do this, we re-train SwinIR-light using the above two training strategies. As shown in Table 5-B of our PDF file (also presented in the Tab. below), UPS delivers superior results compared to both of these optimization methods. Nevertheless, we hope our exploration will inspire future research to develop more effective algorithms to better address this challenge.
>
>
>
> |                             | SwinIR (base)   | SwinIR + Dropout | SwinIR + Pro. Train | UPS
> |-----------------------------|----------------|-------------------|---------------------|---------------------|
> | **PSNR / SSIM**             | 26.47 / 0.7980 | 26.52 / 0.7988    | 26.56 / 0.7986      | 26.83 / 0.8073
> | **Improvement**             |       -        | +0.05 / +0.0008   | +0.09 / +0.0006     | +0.36 / +0.0093
> | **Parameters (K)**     | 930            | 930               | 930                 | 843 (-87)
>
> ## References
>
> [1] Kong, et al. "Reflash dropout in image super-resolution." CVPR, 2022.
>
> [2] Hsu, et al. "DRCT: Saving Image Super-resolution away from Information Bottleneck.", arxiv, 2024.

---

> > ### Comment · Reviewer_fbd3 · 2024-08-13
> > **Thanks**
> >
> > Thanks for the response. Most of the concerns are addressed. I have increased my rating

---

> ### Author Response · Authors · 2024-08-13
> **Thanks for your comments**
>
> Thank you for your kind comments and for appreciating our work. We will include these valuable discussions in the final version of our paper.

---

### Author Rebuttal · Authors · 2024-08-07

Dear AC and Reviewers,

We sincerely thank all the reviewers for their constructive comments and consistent appreciation of the novelty and effectiveness of our UPS for lightweight SISR. Reviewer fbd3, Reviewer RnSV, and Reviewer HE6k have raised concerns about inference efficiency, and both Reviewer RnSV and Reviewer HE6k have shown interest in extending UPS to other SISR scenarios, such as real-world SR or common SISR tasks. Therefore, we address these two issues here.

**Q1. The inference efficiency (latency)**

Thank you for this valuable comment. Results of inference time (ms), FLOPs (G) and GPU memory usage (MB). The speed is tested on an NVIDIA GeForce RTX 2080Ti GPU with an input size of 256 × 256 under x2 lightweight SISR. And we follow other works to calculate the FLOPs at an output resolution of 1280 x 720. Moreover, UPS, HAT-light-UPS, DRCT-light-UPS enhance the inference efficiencies compared with their counterparts.

| Method         | Time (ms)  | FLOPs(G) | Memory  |
| ----------------| ----------- | ---------------------- | ---------------------- |
| RFND-L       | 13      | 146  | 1577      |
| LatticeNet | 18      | 170  | 1639      |
| DLGSA-l    | 225      | 170  | 1800      |
| Omni-SR   | 112      | 195  | 1842      |
| SwinIR-light   | 175   | 244  | 2051      |
| **UPS**            | 119      | 163  | 1785      |
| SwinIR-S    | 117      | 107  | 1365      |
| **UPS-S**          | 71      | 91  | 1039      |
| HAT-light | 153      | 102  | 2039      |
| **HAT-light-UPS** | 136      | 91  | 1763      |
| DRCT-light | 92      | 137  | 2330      |
| **DRCT-light-UPS** | 85      | 125  | 1991      |

 More inference cost comparison can be found in Tab. 2(b)/3/4 of the associated rebuttal PDF for both lightweight and large SISR models. When compared to our baseline model SwinIR-light, UPS reduces the overall inference cost by 33% in terms of FLOPs.

Additionally, as shown in Tab. 3/4, UPS can be adapted to other transformers, consistently enhancing their performance while also reducing inference costs. For example, we integrated UPS into HAT, a state-of-the-art SISR model, resulting in HAT-UPS. We can observe that HAT-UPS significantly reduces computational complexity by 3.52M parameters, 95G FLOPs, and 195ms inference time while yielding improved results. Lastly, we also adopt integrated UPS into lightweight DRCT-light, the DRCT-light-UPS reduces the inference costs by 141K/12G/7ms (parameters/FLOPs/speed) compared to its original version DRCT-light.

**Q2.1. UPS for Real-world SR**

Thank you for appreciating the generalization capability of UPS. We agree with you, as UPS is designed to decouple the optimization of feature extraction and similarity modeling, which is orthogonal to specific methods such as SwinIR. Accordingly, we have extensively explored the benefits of UPS for real-world SR and other frameworks, including HAT and DRCT, under both lightweight and parameter-intensive scenarios. Our experiments show that the proposed UPS consistently enhances efficiency and performance across all these settings (real-world SR, lightweight, and SISR tasks).

For real-world super-resolution (SR), as shown in Tab. 2(a) of the PDF file (also the table below), our proposed UPS-GAN outperforms other state-of-the-art GAN-based and even Diffusion-based methods (Reshift NeurIPS 2023[1] and StableSR IJCV 2024[2]) in terms of NIQE, NRQM, and PI metrics, achieving the best quantitative results (5.09/6.84/4.19). This confirms the effectiveness of UPS for real-world SR tasks.

| Metrics | BSRGAN | RealSR |  ResShift |  StableSR  | SwinIR-GAN | UPS-GAN |
|---------|--------|--------|-----------|-----------|------------|---------|
| NIEQ ↓  | 5.66   | 5.83   | 8.37      | 5.24  | 5.49       | **5.09** |
| NRQM ↑  | 6.27   | 6.32   | 4.56      | 6.12      | 6.48   | **6.84** |
| PI ↓    | 4.75   | 4.40| 7.03     | 4.66      | 4.72       | **4.19** |

**Q2.2. UPS for large model**

Additionally, we explored two more transformers (HAT-light [3] and DRCT-light [4]) for lightweight SISR. As shown in Tab. 4, HAT-light-UPS and DRCT-light-UPS consistently improved their results while achieving better inference latency and lower FLOPs. For instance, DRCT-light-UPS enhances DRCT-light by 0.4dB on the Manga109 dataset and HAT-light-UPS improves HAT-light by 0.28dB on the Urban100 benchmark.

Lastly, we incorporated UPS into large SISR models including SwinIR, HAT [3], and DRCT [4]. The results in Tab. 3 suggest that their enhanced versions by UPS can achieve promising performance and significantly lower inference costs. We can observe that DRCT-UPS surpasses its baseline counterpart (DRCT) by 0.26dB on Urban100.

Thank you again for this constructive suggestion. We will include these analyses in our revised paper. Besides extending our work to image JPEG compression removal and image de-noising tasks, we believe all these comprehensive additional analyses (wide-range SR tasks and easy adaption for other scalable SOTA transformer frameworks) will enhance the value of UPS and inspire future research in developing more effective decoupling optimization algorithms.

[1] Yue, et al. ""Resshift: Efficient diffusion model for image super-resolution by residual shifting." NeurIPS, 2023"

[2] Wang et al. "Exploiting Diffusion Prior for Real-World Image Super-Resolution." IJCV, 2024.

[3] Chen, et al. “Activating More Pixels in Image Super-Resolution Transformer.” CVPR, 2022.

[4] Hsu, et al. “DRCT: Saving Image Super-resolution away from Information Bottleneck.” arxiv, 2024.

---

### Decision · Program_Chairs · 2024-09-25

**Decision:**

Accept (poster)

**Comment:**

The paper presents a lightweight transformer-based single-image super-resolution model. The key idea is decoupling feature extraction and similarity modeling with Unified Projection Sharing. The reviews are mixed, with one Accept, one Borderline accept, and two borderline reject. The general weakness raised by the reviewers are the lack of latency comparison, real-world SR results, and comparison with SOTA super-resolution models.

The authors provided a detailed rebuttal responding to these concerns with additional experimental results. More specifically, the authors provided results on latency comparisons, results on the real-world SR evaluation, and a comparison with Omni-SR using the same training dataset. The reviewers who were leaning positive are satisfied with the clarification and additional results. Unfortunately, the two reviewers who initially leaned negatively toward the paper did not fully engage with the newly presented results.

The AC reviewed the new experimental results and believes that the authors have sufficiently addressed the concerns that were initially raised. Thus, the AC recommends to accept.